# Adipose triglyceride lipase protects renal cell endocytosis in a *Drosophila* dietary model of chronic kidney disease

**Aleksandra Lubojemska**[1], **M. Irina Stefana**[1¤], **Sebastian Sorge**[1], **Andrew P. Bailey**[1], **Lena Lampe**[1], **Azumi Yoshimura**[2], **Alana Burrell**[2], **Lucy Collinson**[2], **Alex P. Gould**[1]*

**1** Physiology and Metabolism Laboratory, The Francis Crick Institute, London, United Kingdom, **2** Electron Microscopy Science Technology Platform, The Francis Crick Institute, London, United Kingdom

¤ Current address: Wellcome Centre for Human Genetics, University of Oxford, Oxford, United Kingdom
* Alex.Gould@crick.ac.uk

**Data Availability Statement:** All relevant data are within the paper and its Supporting Information files. Statistical analysis values are provided in S1

## Abstract

Obesity-related renal lipotoxicity and chronic kidney disease (CKD) are prevalent pathologies with complex aetiologies. One hallmark of renal lipotoxicity is the ectopic accumulation of lipid droplets in kidney podocytes and in proximal tubule cells. Renal lipid droplets are observed in human CKD patients and in high-fat diet (HFD) rodent models, but their precise role remains unclear. Here, we establish a HFD model in *Drosophila* that recapitulates renal lipid droplets and several other aspects of mammalian CKD. Cell type–specific genetic manipulations show that lipid can overflow from adipose tissue and is taken up by renal cells called nephrocytes. A HFD drives nephrocyte lipid uptake via the multiligand receptor Cubilin (Cubn), leading to the ectopic accumulation of lipid droplets. These nephrocyte lipid droplets correlate with endoplasmic reticulum (ER) and mitochondrial deficits, as well as with impaired macromolecular endocytosis, a key conserved function of renal cells. Nephrocyte knockdown of diglyceride acyltransferase 1 (DGAT1), overexpression of adipose triglyceride lipase (ATGL), and epistasis tests together reveal that fatty acid flux through the lipid droplet triglyceride compartment protects the ER, mitochondria, and endocytosis of renal cells. Strikingly, boosting nephrocyte expression of the lipid droplet resident enzyme ATGL is sufficient to rescue HFD-induced defects in renal endocytosis. Moreover, endocytic rescue requires a conserved mitochondrial regulator, peroxisome proliferator-activated receptor-gamma coactivator 1α (PGC1α). This study demonstrates that lipid droplet lipolysis counteracts the harmful effects of a HFD via a mitochondrial pathway that protects renal endocytosis. It also provides a genetic strategy for determining whether lipid droplets in different biological contexts function primarily to release beneficial or to sequester toxic lipids.

## Introduction

In diabetic patients, hyperglycemia triggers complex hemodynamic, metabolic, and inflammatory changes that can lead to a constellation of renal dysfunctions termed "diabetic nephropathy" [1,2]. Obesity is a major risk factor for type 2 diabetes, and it is thought that once adipose

Data and the source data for all main and supporting graphs are provided in S2 Data.

**Funding:** This work was supported by an Investigator Award to APG from the Wellcome Trust (104566/Z/14/Z, https://wellcome.org) and by funding to APG from the Francis Crick Institute (https://www.crick.ac.uk), which receives its core funding from Cancer Research UK (FC001088), the UK Medical Research Council (FC001088) and the Wellcome Trust (FC001088). The funders had no role in study design, data collection and analysis, decision to publish, or preparation of the manuscript.

**Competing interests:** The authors have declared that no competing interests exist.

**Abbreviations:** ATGL, adipose triglyceride lipase; CKD, chronic kidney disease; CLEM, correlative light electron microscopy; CNS, central nervous system; Cubn, Cubilin; DGAT1, diglyceride acyltransferase 1; EI, electron ionization; EMM, estimated marginal mean; ER, endoplasmic reticulum; FAME, fatty acid methyl-ester; GABPA, GA Binding Protein Transcription Factor Subunit Alpha; GC–MS, gas chromatography–mass spectrometry; GLMM, general linear mixed model; HFD, high-fat diet; LDAH, lipid droplet–associated hydrolase; LM, linear model; LMM, linear mixed model; NGS, normal goat serum; PB, phosphate buffer; PBT, PBS + Triton; PGC1α, peroxisome proliferator-activated receptor-gamma coactivator 1α; PQQ, pyrroloquinoline quinone; REML, restricted maximum likelihood; RNAi, RNA interference; ROS, reactive oxygen species; RT, room temperature; SBF SEM, serial blockface scanning electron microscopy; Srl, Spargel; STD, standard diet.

tissue has expanded to its maximum storage capacity, excess lipids then overflow into the bloodstream and trigger lipotoxicity in the kidney and in other peripheral tissues [3–6]. Several mechanisms are thought to contribute to renal lipotoxicity and chronic kidney disease (CKD). For example, the adipo-renal axis is deregulated such that an altered blend of adipokines and other adipose-derived factors produces renal inflammation, fibrosis, and oxidative stress, leading to defective glomerular filtration and proteinuria [7,8]. Adipose-derived factors as well as ectopic lipid accumulation in the kidney are thought to impact multiple podocyte, endothelial and proximal tubule functions, at least in part, via the promotion of renal insulin resistance [9]. Rodent studies using a high-fat diet (HFD) have provided valuable insights into the links between lipotoxicity and CKD. In mice, HFD is sufficient to trigger features of stress and damage in mouse proximal tubule cells, including the endoplasmic reticulum (ER) unfolded protein response, lipid peroxidation, and defective albumin reabsorption [10–13]. In both human patients and mouse models, mitochondrial loss and dysfunction are central to the development and progression of CKD [14]. The underlying abnormalities include decreased mitochondrial biogenesis, loss of mitochondrial membrane potential, decreased ATP generation, and altered levels of reactive oxygen species (ROS).

One hallmark of CKD lipotoxicity is the accumulation of lipid droplets in podocytes and in proximal tubule epithelial cells. Lipid droplets are intracellular organelles comprising a core of neutral lipids, such as triglycerides, surrounded by a polar lipid monolayer containing many different proteins, some of which function in lipid metabolism [15,16]. Nascent lipid droplets form via a complex process involving neutral lipid synthesis by ER enzymes such as diglyceride acyltransferase 1 (DGAT1) [17,18]. The neutral lipids stored in lipid droplets can then be broken down by lipolysis, mediated via lipid droplet-associated enzymes such as adipose triglyceride lipase (ATGL) [19]. This catabolic process is distinct from lipophagy, which involves lysosomal acid lipase acting upon lipids delivered to autolysosomes via autophagy [20]. As early as the 1930s, lipid droplets were observed as a sign of pathology in podocytes and in proximal tubule cells during renal disease [21,22], yet it has remained unclear whether they play a protective or a harmful role. Resolution of this question is important for understanding CKD mechanisms and will likely require in vivo studies of HFD animal models with cell type–specific manipulations of enzymes that directly regulate the neutral lipid cargo of droplets rather than acting on other aspects of fatty acid metabolism. ATGL is of particular interest here as its specific function in renal cells in HFD and other CKD mouse models is not yet clear, although whole-body knockouts fed a standard diet (STD) are known to display proximal tubule damage and podocyte apoptosis [23,24].

The animal model *Drosophila* has powerful genetics for studying the molecular pathogenesis of some human diseases. This approach is possible because of extensive physiological similarities between major fly and human organs, including the kidney [25]. In *Drosophila*, the renal system comprises 2 anatomically distinct components—Malpighian tubules and nephrocytes [26–28] (**Fig 1A**). Malpighian tubules are excretory cells that also function in salt and water balance, similar to mammalian renal tubules [29,30]. Nephrocytes are podocyte-like cells that regulate the composition of the hemolymph (blood) via a filtration barrier consisting of Sns and Kirre, orthologs of the mammalian slit diaphragm proteins Nephrin and Neph1 [31,32]. Nephrocytes also function like mammalian proximal tubule cells, efficiently reabsorbing macromolecules via a Cubilin (Cubn)-dependent endocytic receptor complex [33,34]. *Drosophila* has thus been used to model several monogenic kidney diseases including steroid-resistant nephrotic syndrome and renal Fanconi syndrome [34,35]. Previous work using *Drosophila* has also shown that chronic high dietary sugar during adulthood increases O-GlcNAcylation, in turn leading to decreased Sns expression and compromised nephrocyte function [36]. Here, we establish a *Drosophila* HFD model that recapitulates in nephrocytes the ectopic

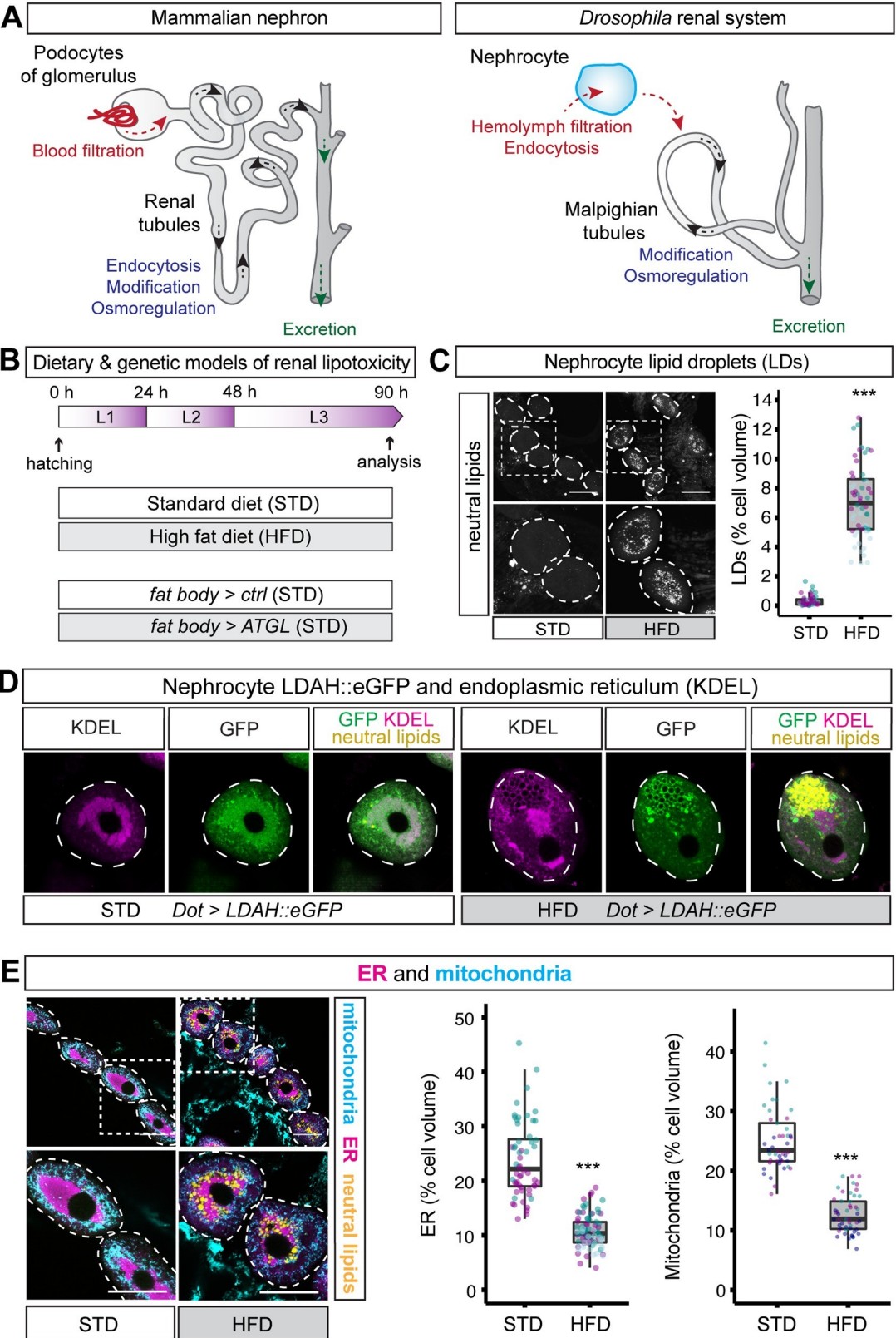

**Fig 1. HFD induces LDs and decreases ER and mitochondria in nephrocytes.** (**A**) Diagram comparing the mammalian nephron and the *Drosophila* renal system. *Drosophila* nephrocytes share functions with mammalian podocytes of the glomerulus

and also with proximal tubules. *Drosophila* Malpighian tubules are functionally analogous to renal tubules. (**B**) For the chronic dietary model, larvae are fed either a STD or a HFD throughout development. The chronic genetic model for lipid overflow on STD utilises a fat body–specific GAL4 driver (*Lpp-GAL4*) to compare fat body overexpression of ATGL (*Lpp>ATGL*) with the control genotype (*Lpp>ctrl*). (**C**) LDs, stained with a neutral lipid dye (LipidTOX), are more abundant in pericardial nephrocytes (dashed outlines) of HFD than STD larvae. Graph quantifies % of nephrocyte volume occupied by LDs in STD and HFD larvae. In this and subsequent graphs, the boxplot encompasses the first to third quartile and shows the median, and whiskers extend from the hinge by 1.5× interquartile range. Data points are coloured according to which independent experiment they are from. Data were statistically analysed using LMMs followed by a Wald chi-squared test. Asterisks show statistical significance (* $p < 0.05$, ** $p < 0.005$, *** $p < 0.0005$), and ns indicates $p > 0.05$ in this and all subsequent graphs. See **S1 Data** for details of *p*-values and the type of statistical model used for all graphs in this study. **S2 Data** provides the source data used for all graphs and statistical analyses. Scale bar = 50 μm. (**D**) In pericardial nephrocytes (dashed outlines), LDAH::GFP localises primarily to the ER (marked with anti-KDEL antibody) in STD larvae but to the surface of ER-associated LDs (marked with the neutral lipid dye LipidTOX) in HFD larvae. Note that LDAH::GFP induces clustering of LDs [38]. (**E**) Low and high magnification views of pericardial nephrocytes (dotted outlines) from STD and HFD larvae, showing that HFD decreases mitochondria (marked with anti-ATP5A) and ER (marked with anti-KDEL) but increases LDs (marked with LipidTOX). Quantitations of ER and mitochondrial volumes are shown as a % of nephrocyte volume in STD and HFD larvae. ATGL, adipose triglyceride lipase; ER, endoplasmic reticulum; HFD, high-fat diet; LD, lipid droplet; LDAH, lipid droplet–associated hydrolase; LMM, linear mixed model; STD, standard diet.

lipid droplets and cellular dysfunction observed in CKD. This CKD model is then interrogated with cell type–specific genetics and assays for mitochondria and endocytic function to pinpoint the role of lipid droplets in renal lipotoxicity. Genetic rescues and other approaches are then used to test whether lipid droplet enzymes are necessary and sufficient to ameliorate multiple aspects of renal dysfunction induced by HFD exposure.

## Results

### HFD induces lipid droplets and abnormal nephrocyte ER and mitochondria

We established a *Drosophila* model for diet-induced renal lipotoxicity by raising animals on a HFD throughout larval development (0 to 90 hours after hatching; see Materials and methods) (**Fig 1B**). Gas chromatography–mass spectrometry (GC–MS) measurements of total fatty acids in larval hemolymph showed that chronic exposure to HFD led to a large increase in circulating oleate (C18:1) and myristoleate (C14:1) but not myristate (C14:0) (**S1A Fig**). HFD is supplemented with oleic acid suggesting that the hemolymph increase of this fatty acid derives directly from the diet. High myristoleate may therefore reflect conversion from dietary oleate and is consistent with previous observations that the most abundant fatty acids in the hemolymph have an average chain length of 12 to 14 carbons [37].

Compared to STD, HFD did not significantly alter body growth or developmental timing, but it did lead to a small increase in the size of nephrocytes (**S1B–S1D Fig**). To begin characterising the effects of HFD on nephrocytes, a neutral lipid dye (LipidTOX) was used to reveal that lipid droplets in pericardial nephrocytes are sparse in STD animals but strikingly abundant in HFD animals (**Fig 1C**). GFP fused to lipid droplet–associated hydrolase (LDAH) localises to the ER and to the surface of lipid droplets [38]. In STD animals, *Dot-GAL4*–driven expression of LDAH (*Dot>LDAH::GFP*) specifically in nephrocytes was mostly ER associated but, in HFD larvae, it predominantly localised to the surface of 1- to 2-μm diameter lipid droplets that stain strongly with the neutral lipid dye (**Fig 1D**). Hence, chronic exposure to HFD leads to the strong accumulation of nephrocyte lipid droplets. We also observed that HFD markedly decreased the overall volume of ER and mitochondria in nephrocytes, approximately halving the proportion of the total cell volume that each organelle occupies (**Fig 1E**). These observations together show that HFD in *Drosophila*, as in mammals, induces renal lipid droplets and also a deficit in ER and mitochondrial volumes.

## HFD compromises nephrocyte endocytosis

An important function of nephrocytes is to resorb circulating proteins and other macromolecules from the hemolymph via Cubn- and Amnionless-dependent endocytosis [31,33]. This nephrocyte endocytic function can be quantified by monitoring ex vivo uptake of the polysaccharide dextran [31]. A combination of fluorescently labelled 10 kDa and 500 kDa dextrans has previously been used to assess size-selective filtration as well as overall endocytosis [31]. Using this approach, we measured mean dextran intensities in nephrocytes but observed only a modest increase in the 500:10 kDa dextran intensity ratio over an ex vivo incubation time course of 3 to 20 minutes (S2 Fig). A 30-minute ex vivo incubation time was therefore subsequently used as a robust readout for nephrocyte endocytosis rather than size-selective filtration. This assay revealed that endocytic uptake of dextran is decreased in the nephrocytes of HFD animals and, although there is cell-to-cell variability, the mean overall reduction is approximately 50% compared with STD animals (Fig 2A and 2B). This finding is further strengthened by a modified ex vivo nephrocyte uptake assay that utilised labelled albumin. As with dextran, nephrocyte accumulation of albumin was significantly decreased by HFD (Fig 2C and 2D). We therefore conclude that HFD compromises the key renal function of nephrocyte endocytosis.

To determine the ultrastructural changes associated with HFD compromised endocytosis, we used correlative light electron microscopy (CLEM) with Airyscan confocal microscopy and serial blockface scanning electron microscopy (SBF SEM). The plasma membrane of nephrocytes is organised into a dense undulating network of slit diaphragms and lacunae [31,32,39]. These ultrastructural features of the plasma membrane network are visible with SBF SEM in both STD and HFD nephrocytes (Fig 3A). CLEM analysis of nephrocytes carrying the endogenous *Rab7* gene tagged with *YFP^myc*, *Rab7*::*YFP^myc* [YRab7,40], distinguished 5 endolysosomal compartments according to the "white," "light," or "dark" SEM luminal density and the Dextran and Rab7 labelling status (Fig 3A, S3 Fig). Comparing our CLEM analysis with previous nephrocyte studies [41–43] strongly suggested that the "white" compartment corresponds to a

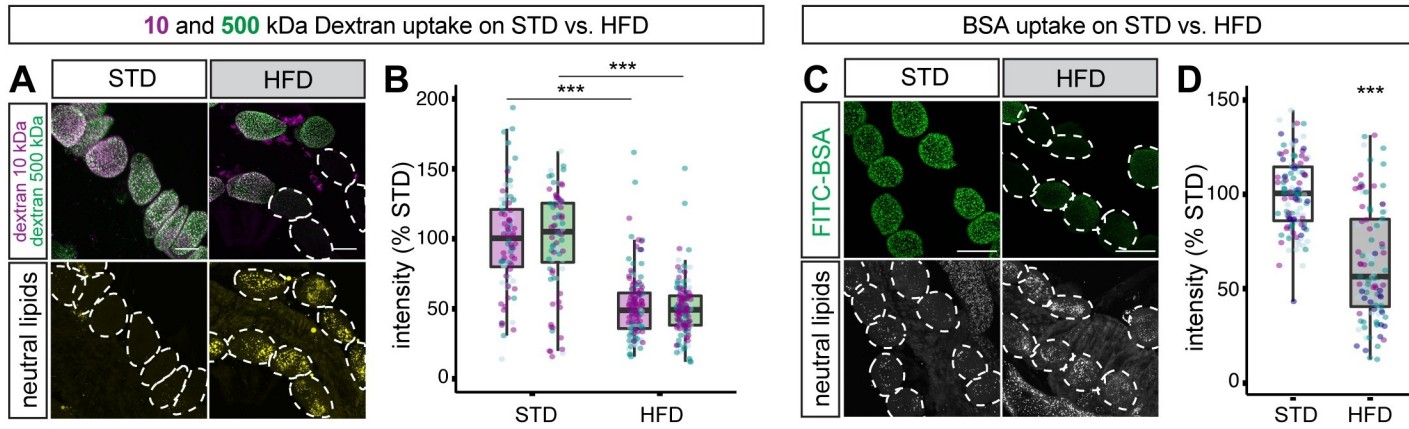

**Fig 2. HFD decreases nephrocyte uptake of dextran and albumin.** (**A**, **B**) Dextran uptake assay. (A) Nephrocytes from STD and HFD larvae shown after ex vivo incubation with labelled 10-kDa (magenta) and 500-kDa (green) dextran. Bottom row shows same field of view with lipid droplets revealed with a neutral lipid stain (LipidTOX). Dashed outlines indicate the positions of all nephrocytes in the bottom row, but only those that show weak dextran uptake in the top row. (B) Graph shows that uptake of both 10-kDa (magenta) and 500-kDa (green) dextran is significantly higher on STD than on HFD (*p* < 0.0005). Scale bar = 30 μm. (**C**, **D**) Albumin uptake assay. (C) Nephrocytes from STD and HFD larvae shown after ex vivo incubation with labelled bovine serum albumin (FITC-BSA). Bottom row shows same field of view with lipid droplets revealed with a neutral lipid stain (LipidTOX). Dashed outlines indicate the positions of all nephrocytes in the bottom row, but only those that show weak albumin uptake in the top row. (D) Graph shows that uptake of albumin is significantly higher on STD than HFD (*p* < 0.0005). Scale bar = 50 μm. See S1 Data for details of *p*-values and the type of statistical model used for all graphs in this study. S2 Data provides the source data used for all graphs and statistical analyses. HFD, high-fat diet; STD, standard diet.

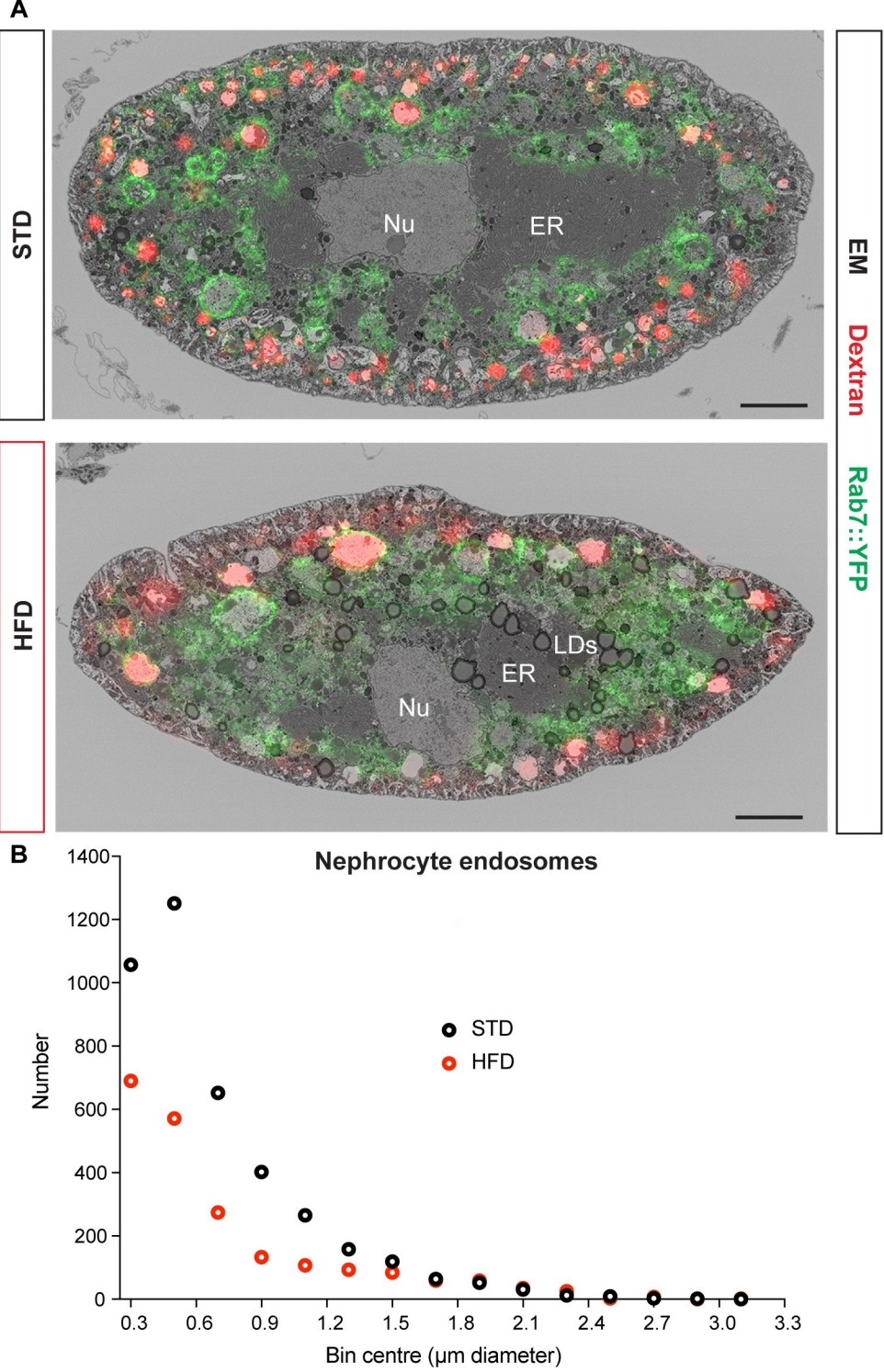

**Fig 3. HFD decreases the number of nephrocyte endosomes.** (**A**) CLEM images of midsections of STD and HFD nephrocytes expressing Rab7::YFP[myc] and labelled with Alexa Fluor 568 10 kDa dextran. Scale bars are 5 μm. (**B**) Distribution of endosome number versus diameter for 1 STD and 1 HFD nephrocyte segmented by SBF SEM according to the "white" and "light" classifications in **S3A Fig**. See **S1 Data** for details of *p*-values and the type of statistical model used for all graphs in this study. **S2 Data** provides the source data used for all graphs and statistical analyses. CLEM, correlative light electron microscopy; ER, endoplasmic reticulum; HFD, high-fat diet; LD, lipid droplet; SBF SEM, serial blockface scanning electron microscopy; STD, standard diet.

mix of Dextran$^+$Rab7$^-$ early endosomes and Dextran$^+$Rab7$^+$ endosomes (alpha-vacuoles). The "light" compartment encompasses Dextran$^+$Rab7$^+$ endosomes and Dextran$^-$Rab7$^+$ late endosomes (beta-vacuoles), whereas the "dark" compartment included both Dextran$^-$Rab7$^+$ late endosomes and Dextran$^-$Rab7$^-$ lysosomes. Based on this CLEM classification, the "white" and "light" compartments were segmented from the SBF SEM stacks of entire nephrocyte cells to provide the size distributions of endosomes. This segmentation approach revealed that HFD nephrocytes have substantially fewer endosomes than STD nephrocytes (**Fig 3B**). This HFD deficit is particularly striking for endosomes of less than 1 μm in diameter, and it is likely to account for the observed decrease in the capacity of nephrocytes to uptake macromolecules such as dextran.

We have provided evidence demonstrating that HFD induces a syndrome of nephrocyte abnormalities including induction of lipid droplets and deficits in the ER, mitochondria, and endocytic compartment. Many of these abnormalities are strikingly similar to those observed in the proximal tubule cells of HFD mice and CKD patients. This establishes the *Drosophila* HFD paradigm as a useful model for kidney disease. We next combined our new animal model with cell type–specific genetic manipulations in order to identify the mechanisms linking high dietary lipid to nephrocyte dysfunction. In particular, we focused on the role of lipid droplets, determining whether they are beneficial or harmful for renal function.

## Renal lipid droplets can be induced via adipose tissue lipolysis and blocked via Cubilin-dependent endocytosis

To define the physiological pathway from HFD to elevated hemolymph fatty acids and then to nephrocyte lipid droplets, we directly tested the role of lipid overflow from the larval *Drosophila* adipose tissue (fat body) to peripheral tissues [44]. *Lpp-GAL4* was used to drive chronic expression of the ATGL orthologue Brummer [45] in the fat body (*Lpp>ATGL*) (**Fig 1B**). As with HFD, fat body–specific ATGL expression in STD animals did not substantially alter growth or developmental timing, although it did decrease nephrocyte volume by approximately 25% (**S1E–S1G Fig**). Importantly, this genetic manipulation was sufficient to induce robust lipid droplet accumulation in nephrocytes of STD animals, suggesting that lipid overflow from adipose tissue may also be relevant for HFD-induced renal lipid droplets (**Fig 4A and 4B**). Lipid overflow from adipose tissue, like HFD, also lead to a functional deficit in nephrocyte endocytosis, as *Lpp>ATGL* animals also showed impaired dextran uptake (**Fig 4C and 4D**).

In mammalian proximal tubule cells, the Cubn receptor is known to be involved in the endocytic uptake of lipoproteins as well as proteins [46]. *Dot-GAL4* was therefore used to drive RNA interference (RNAi) for *Drosophila* Cubn specifically in nephrocytes (*Dot>Cubn[i]*). This revealed that, on HFD, *Cubn* is required for the accumulation of nephrocyte lipid droplets (**Fig 4E and 4F**). With the preceding results, this provides evidence supporting the conclusion that HFD leads to excess fat circulating in the hemolymph (blood), which is then endocytosed by nephrocytes via the Cubn receptor and targeted to lipid droplets. Given that Cubn knockdown did not significantly decrease nephrocyte uptake of a labelled free fatty acid (BODIPY FL C12), it is likely that lipoproteins are the major form of circulating fat that contributes to nephrocyte lipid droplets (**Fig 4G and 4H**).

## Boosting ATGL expression rescues HFD-induced nephrocyte dysfunction

Our results show that excess circulating lipids are endocytosed by nephrocytes and targeted to lipid droplets. This raises an important question—what, if any, contribution do lipid droplets make towards HFD-induced renal dysfunction? To identify unambiguous functions for lipid

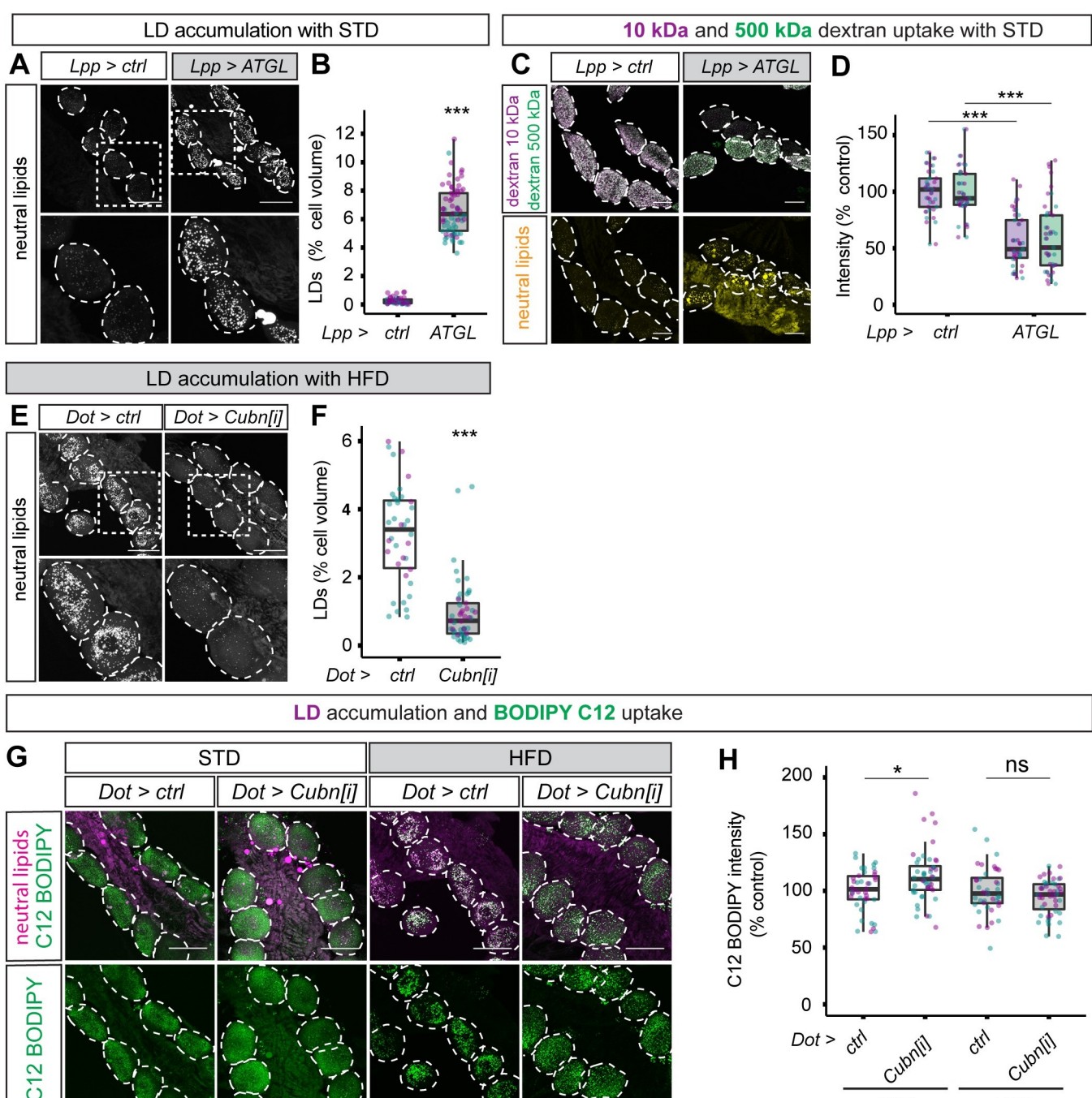

**Fig 4. Fat body lipolysis induces and Cubn-dependent endocytosis blocks renal lipid droplets.** (**A**, **B**) Confocal panels (A) and quantifications (B) show that lipid droplets (marked with LipidTOX) accumulate in nephrocytes (dashed outlines in A) of STD larvae expressing ATGL in the fat body (*Lpp>ATGL*) but not in controls (*Lpp>ctrl*). (**C**, **D**) Confocal micrographs (C) and quantifications of dextran mean fluorescence intensity (D) show that ex vivo dextran uptake is decreased in nephrocytes of larvae expressing ATGL in the fat body (*Lpp>ATGL*) but not in those of control larvae (*Lpp>ctrl*). Similar results are obtained with 10-kDa and 500-kDa dextrans. (**E**, **F**) Confocal micrographs (E) and quantifications (F) show that lipid droplets (marked with LipidTOX) are far less abundant in the nephrocytes (dashed outlines in E) of nephrocyte-specific Cubn knockdown (*Dot>Cubn[i]*) compared to control (*Dot>ctrl*) HFD larvae. (**G**, **H**) Confocal micrographs (G) and quantifications of fluorescence intensity (H) show that Cubn knockdown in nephrocytes (*Dot>Cubn[i]*) does not decrease ex vivo uptake of a fluorescent analogue of a C18 saturated free fatty acid (BODIPY FL C12) by nephrocytes from STD or HFD larvae. Panel G shows BODIPY FL C12 (green) and LipidTOX (magenta). See **S1 Data** for details of *p*-values and the type of statistical model used for all graphs in this study. **S2 Data** provides the source data used for all graphs and statistical analyses. ATGL, adipose triglyceride lipase; Cubn, Cubilin; HFD, high-fat diet; LD, lipid droplet; STD, standard diet.

droplets, rather than for fatty acid metabolism more generally, we targeted 2 enzymes with direct substrates/products corresponding to the triglyceride cargo of droplets, DGAT1/Midway and ATGL/Brummer. Importantly, nephrocyte lipid droplets in HFD animals were efficiently inhibited either by knocking down DGAT1 (*Dot>DGAT1[i]*) or by increasing the expression of ATGL (*Dot>ATGL*) (**Fig 5A**). Systematically comparing the HFD phenotypes of these 2 genetic manipulations allows the roles of lipid droplet triglycerides to be parsed into synthesis versus lipolysis functions. This comparative strategy revealed that DGAT1 knockdown in HFD nephrocytes gave small or nonsignificant changes in mitochondrial and ER volumes, respectively (**Fig 5B and 5C**). Blocking lipid droplets via ATGL expression, however, did significantly increase both mitochondrial and ER volumes in HFD nephrocytes, consistent with partial restoration of these cell parameters towards STD values (**Fig 5B and 5C**, compare with **Fig 1E**). Using the ratiometric dye BODIPY 581/591 C11 to detect lipid peroxidation, we observed no difference between STD and HFD nephrocytes (**S4 Fig**). Furthermore, lipid peroxidation on HFD did not significantly change with ATGL expression, but it was strongly elevated with DGAT1 knockdown (**S4 Fig**). Together, these results demonstrate that abrogation of lipid droplets in HFD nephrocytes by increasing ATGL lipolysis is able to rescue significantly the mitochondrial and ER volumes without increasing lipid peroxidation. In contrast, blocking lipid droplet biogenesis in HFD nephrocytes via inactivation of DGAT1 triglyceride synthesis fails to rescue mitochondria and ER and also increases potentially cytotoxic lipid peroxidation.

We next assessed nephrocyte endocytic function. SBF SEM was used to analyse the entire cell volumes of STD, HFD, HFD DGAT1[i], and HFD ATGL nephrocytes (**S1–S4 Movies**). Using our CLEM classification to segment these 4 nephrocyte volumes revealed that the HFD-associated decrease in the total number and volume of endosomes is fully rescued by ATGL but not by DGAT1[i] (**Fig 5D and 5E**). In line with this, both the dextran and albumin uptake of HFD nephrocytes were completely rescued by ATGL expression but not by DGAT1 knockdown (**Fig 5F and 5G**). Importantly, DGAT1 knockdown was epistatic to ATGL expression with respect to nephrocyte dextran uptake (**Fig 5H**). Hence, ATGL protects renal endocytic function via a mechanism requiring triglyceride substrates, rather than by any moonlighting activity of the enzyme. Together, these striking findings show that nephrocyte-specific ATGL expression is sufficient to ameliorate HFD-induced mitochondrial defects and to stimulate full rescue of endocytic function.

## ATGL rescue of HFD nephrocyte function requires Srl and Delg

We reasoned that *UAS-ATGL* rescue of nephrocyte dysfunction could reflect restoration of HFD-induced down-regulation of the endogenous *bmm/ATGL* gene, which has been reported in whole adult flies [47]. To test this possibility, we monitored *ATGL* gene expression using a *bmm-GFP* transcriptional reporter (*ATGL-GFP*) [48]. This approach revealed that HFD leads to a significant decrease in *ATGL* reporter expression (**Fig 6A and 6B**). Together with the ATGL rescue experiments, this suggests that down-regulation of *ATGL* expression could contribute to nephrocyte dysfunction on HFD. Interestingly, *ATGL-GFP* expression in HFD nephrocytes was restored to approximately STD levels by providing exogenous ATGL enzyme (*Dot>ATGL*) (**Fig 6C**). Thus, ATGL in nephrocytes not only regulates mitochondria but may also regulate the transcription of its own gene, raising the question of whether these 2 ATGL functions are separate or linked. To address this, we manipulated peroxisome proliferator-activated receptor-gamma coactivator 1α (PGC1α), a transcriptional coactivator that controls mitochondrial biogenesis and energy metabolism, also mediating proximal tubule recovery from kidney disease [49,50]. Spargel (Srl), the *Drosophila* PGC1α ortholog, regulates

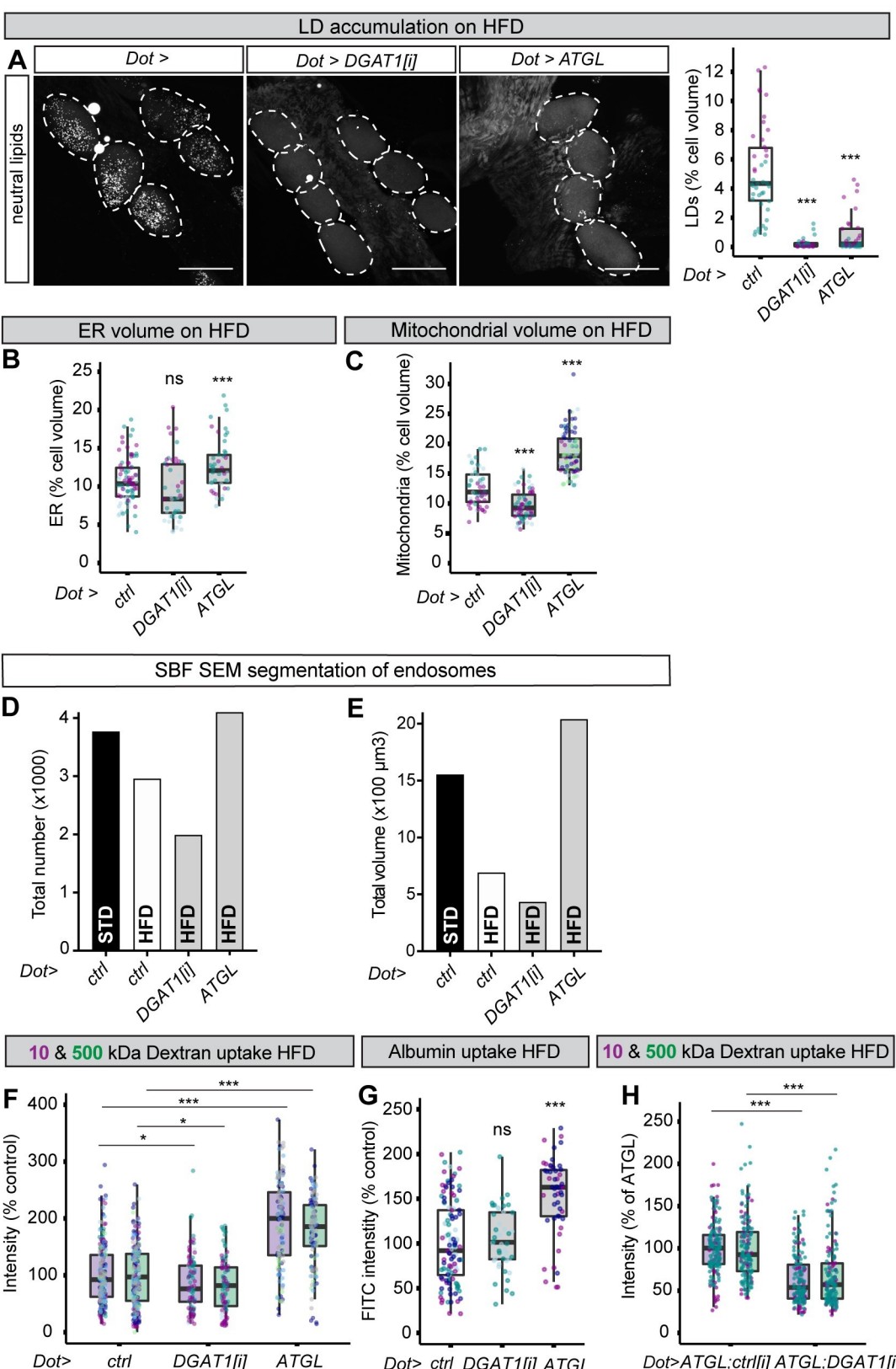

**Fig 5. ATGL rescues mitochondria and macromolecule uptake of HFD nephrocytes.** (**A**) Confocal micrographs of nephrocytes (dotted outlines), stained with a dye for neutral lipids (LipidTOX), and corresponding quantifications showing that LD accumulation observed on HFD in control larvae (*Dot>ctrl*) is almost completely blocked in nephrocyte-specific DGAT1 RNAi (*Dot>DGAT1[i]*) or ATGL expression (*Dot>ATGL*) larvae. Scale bar = 50 μm. (**B**, **C**) Quantifications of nephrocyte ER volume (B) and mitochondrial volume (C) for HFD larvae of the control (*Dot>ctrl)*, *Dot>DGAT1[i]* and *Dot>ATGL* genotypes. Note that the difference in mitochondrial volumes between *Dot>ctrl* and *Dot>DGAT1[i]*, although significant in this figure, is small in magnitude and is not significant in **Fig 6E**. (**D**, **E**) SBF SEM quantifications of total endosome numbers (D) and volumes (E) in STD control (*Dot>ctrl*), HFD control (*Dot>ctrl*), HFD *Dot>DGAT1[i]*, and HFD *Dot>ATGL* nephrocytes. (**F**, **G**) Quantifications of 10-kDa and 500-kDa dextran uptake (F) and albumin (FITC-BSA) uptake (G) in HFD nephrocytes showing that nephrocyte-specific ATGL expression (*Dot>ATGL*), but not DGAT1 knockdown (*Dot>DGAT1[i]*), rescues HFD nephrocyte endocytic function. (**H**) Quantifications of nephrocyte uptake of 10-kDa and 500-kDa dextrans showing that the genetic rescue of HFD nephrocyte endocytic function obtained with nephrocyte-specific ATGL overexpression (*Dot>ATGL; ctrl[i]*) is blocked by concomitant DGAT1 knockdown (*Dot>ATGL; DGAT1[i]*). *ctrl[i]* refers to *UAS-mCherry[i]*. See **S1 Data** for details of *p*-values and the type of statistical model used for all graphs in this study. **S2 Data** provides the source data used for all graphs and statistical analyses. ATGL, adipose triglyceride lipase; DGAT1, diglyceride acyltransferase 1; ER, endoplasmic reticulum; HFD, high-fat diet; LD, lipid droplet; RNAi, RNA interference; SBF SEM, serial blockface scanning electron microscopy; STD, standard diet.

mitochondrial activity, and it functions redundantly with the GA Binding Protein Transcription Factor Subunit Alpha (GABPA) ortholog Ets97D/Delg to promote mitochondrial biogenesis [51,52]. Furthermore, Srl overexpression is known to counteract HFD-induced dysfunction of the *Drosophila* heart [47]. Using nephrocyte-specific RNAi knockdowns, we found that *Srl* and *Delg* are each required for the normal mitochondrial volume of STD nephrocytes (**S5A Fig**). Importantly, knockdown of *PGC1α/Srl* also decreased *ATGL-GFP* expression in STD nephrocytes (**Fig 6D**). This finding suggests that a key transcriptional coactivator of mitochondrial genes, PGC1α, is required directly or indirectly to regulate the expression of the *ATGL* gene.

To define the role of PGC1α/Srl in HFD nephrocyte dysfunction, we used both pharmacological and genetic approaches. Pyrroloquinoline quinone (PQQ), an indirect activator of PGC1α/Srl [53,54], was able to restore substantially the mitochondrial volume of control or DGAT1[i] HFD nephrocytes (**Fig 6E**). Strikingly, the degree of rescue of HFD mitochondrial volume with PQQ was comparable to that achieved via ATGL expression (**Fig 6E**). The PQQ experiments rule out that PGC1α solely acts upstream of DGAT1-dependent triglyceride biosynthesis and, together with the *GFP* reporter analysis, suggest that a HFD-induced decrease in PGC1α expression/activity could down-regulate *ATGL* gene expression. Importantly, genetic knockdown of *PGC1α/Srl*, or *Delg*, inhibited ATGL rescue of HFD nephrocyte mitochondrial volume (**Fig 6F**). *Srl* knockdown also completely blocked ATGL rescue of nephrocyte dextran uptake, which remained at or slightly below the level that is observed in control genotype HFD animals (**Fig 6G**, **S5B Fig**). These pharmacological and genetic experiments together demonstrate that PGC1α is required for the ATGL rescue of HFD-induced deficits in nephrocyte mitochondria and endocytosis.

## Discussion

This study establishes the first *Drosophila* model for HFD-induced CKD. Our results reveal that exposure to HFD elevates circulating fatty acids and induces renal defects in *Drosophila* that are strikingly similar to those observed in mammals. Key metabolic features of CKD in podocytes and proximal tubule cells are recapitulated in *Drosophila* nephrocytes including lipid droplet induction, a decrease in mitochondrial volume, as well as compromised endocytic uptake of albumin and other macromolecules. The powerful genetics and high-throughput possibilities of the *Drosophila* model open up a significant new avenue for in vivo mechanistic studies of CKD. We now discuss the mechanism by which HFD induces CKD-like dysfunction in *Drosophila* and how increased ATGL/Bmm expression rescues it. We also discuss how side-

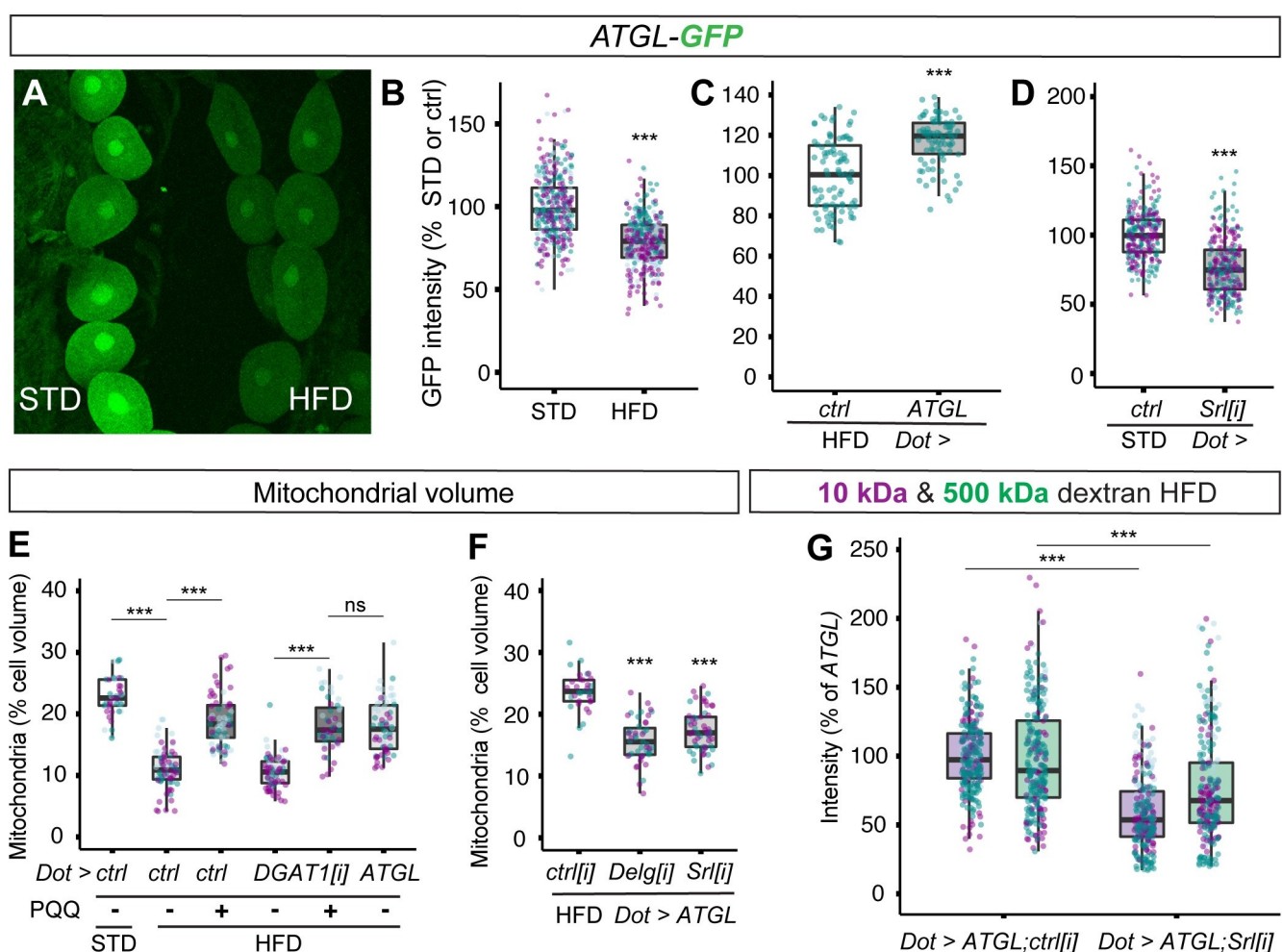

**Fig 6. Rescue of nephrocyte mitochondria and endocytosis on HFD requires PGC1α/Srl. (A–D)** Confocal micrographs of nephrocytes and quantifications showing that *ATGL-GFP* reporter expression is suppressed by HFD (A, B), restored by *Dot>ATGL* (C), and decreased by *Dot>Srl[i]* on STD diet (D). Note that the GFP intensities of HFD and STD nephrocytes can be directly compared in panel A because they are both imaged within the same field of view. (**E**) Dietary supplementation with PQQ rescues nephrocyte mitochondrial volume on HFD, in control or *Dot>DGAT1[i]* genotypes, back to a value similar to that in HFD *Dot>ATGL* or STD larvae. (**F**) *Dot>ATGL* rescue of nephrocyte mitochondrial volume is compromised by simultaneous knockdown of *PGC1α* (*Dot>ATGL;Srl[i]*) or Delg (*Dot>ATGL;Delg[i]*). *ctrl[i]* refers to *UAS-mCherry[i]*. (**G**) *Dot>ATGL* rescue of 10-kDa and 500-kDa dextran uptake on HFD is blocked by simultaneous knockdown of *PGC1α* (*Dot>ATGL;Srl[i]*). *ctrl[i]* refers to *UAS-mCherry[i]*. See **S1 Data** for details of *p*-values and the type of statistical model used for all graphs in this study. **S2 Data** provides the source data used for all graphs and statistical analyses. ATGL, adipose triglyceride lipase; DGAT1, diglyceride acyltransferase 1; HFD, high-fat diet; PGC1α, peroxisome proliferator-activated receptor-gamma coactivator 1α; PQQ, pyrroloquinoline quinone; Srl, Spargel; STD, standard diet.

by-side functional comparisons of the triglyceride metabolic enzymes DGAT1 and ATGL provide a widely applicable strategy for clarifying the cellular functions of stress-induced lipid droplets.

## Boosting fatty acid flux through the triglyceride compartment protects renal endocytosis

ATGL overexpression is predicted to increase the release of free fatty acids, a change that is associated with lipotoxicity. Nevertheless, we find that the outcome of this genetic manipulation can either be beneficial or harmful for renal endocytosis, depending upon whether it is adipose or nephrocyte specific. We showed that HFD induction of lipid droplets and endocytic

dysfunction in nephrocytes are both mimicked on STD via overexpression of ATGL in adipose tissue. Furthermore, HFD induction of nephrocyte lipid droplets requires the Cubn endocytic receptor. Together with the finding that HFD elevates hemolymph fatty acids, these results suggest that excess diet-derived fatty acids are mobilised from adipose tissue into the circulation, taken up by nephrocytes via receptor-mediated endocytosis, and subsequently accumulate in the core of lipid droplets.

A key finding of this study is that experimentally boosting the expression of ATGL in nephrocytes rescues most of the deleterious effects of HFD on the morphology and function of these cells. Thus, ATGL expression substantially restored ER volume, mitochondrial volume, endosomal number and, importantly, the endocytosis of dextran and albumin. This striking protection of nephrocyte endocytic function by ATGL is strictly dependent upon DGAT1, strongly suggesting that it requires fatty acid flux into and out of the lipid droplet triglyceride compartment. Our analysis also suggests that fatty acid flux through the triglyceride compartment is suboptimal on HFD because ATGL becomes limiting. The observation that HFD decreases nephrocyte *ATGL* reporter expression is indicative of repression at the transcriptional level. However, our results do not rule out an additional contribution to HFD repression of ATGL from posttranscriptional mechanisms.

## A general strategy for distinguishing between different lipid droplet functions

Our systematic comparisons between 2 different genetic methods for inhibiting stress-induced lipid droplets have important implications for interpreting the role of these organelles in a wide range of different biological contexts. In the case of nephrocytes, we have shown that either blocking the last step of triglyceride synthesis (DGAT1 knockdown) or boosting lipolysis (ATGL overexpression) efficiently prevents the accumulation of lipid droplets, yet these manipulations produce very different functional outcomes. We now outline how side-by-side comparisons of both genetic perturbations can be used to distinguish whether lipid droplets are harmful or protective and, if they are protective, to provide insights into the underlying mechanism. DGAT1 knockdown in HFD nephrocytes increases lipid peroxidation damage and decreases endocytosis. Hence, as for the hypoxic central nervous system (CNS) [55], synthesis of triglycerides has a net protective effect on nephrocytes. DGAT1 could also contribute beneficially by preventing the accumulation of diglycerides, which are known to play important roles as signalling lipids and phospholipid precursors [56]. To determine how the triglyceride core of lipid droplets may be protective, it is also important to consider the functional effect of boosting lipolysis via ATGL. A harmful outcome implies that the triglyceride compartment protects by sequestering potentially toxic lipids, whereas a beneficial effect suggests that it releases protective lipids, with signalling or other roles [19,57,58]. In the case of HFD nephrocytes, we found that boosting ATGL rescues macromolecular uptake, suggesting that nephrocyte lipid droplets protect by acting as a source of beneficial lipids. Similar reasoning suggests a reinterpretation of the Bmm/ATGL experiments in 2 previous studies in the adult *Drosophila* retina, which reported that glial lipid droplets induced either by mitochondrial defects or by loss of ADAM17 metalloprotease act to promote neurodegeneration [59,60]. Both studies found that decreasing lipid droplets via the overexpression of Bmm/ATGL led to less neurodegeneration. Our DGAT1 and ATGL comparisons now suggest that boosting ATGL equates to a gain not a loss of function for lipid droplets or more precisely for their role as a platform for triglyceride lipolysis. Therefore, the previous retinal studies and our own nephrocyte findings appear to concur with respect to showing that the lipolysis function of lipid droplets can be beneficial. These comparisons illustrate that assigning protective or harmful roles to lipid droplets requires a nuanced approach that parses their various subfunctions.

## ATGL rescues renal endocytic dysfunction via the PGC1α pathway

This study reveals that ATGL/Bmm rescues HFD-induced dysfunction in *Drosophila* renal cells via a mechanism that requires GABPA/Delg and also PGC1α/Srl, a conserved regulator of mitochondrial biogenesis, membrane potential, and ß-oxidation [49,50]. A pharmacological approach provided evidence that PGC1α is sufficient to correct HFD deficits in nephrocyte mitochondrial volume. Unlike ATGL rescue, PGC1α rescue does not require DGAT1-dependent triglyceride synthesis. Moreover, genetic epistasis tests demonstrated that PGC1α is necessary for ATGL to rescue both the mitochondrial volume and the endocytic dysfunction of HFD nephrocytes. These findings together make it likely that triglyceride synthesis and ATGL-dependent lipolysis act upstream of the PGC1α-dependent mitochondrial processes required for optimal nephrocyte endocytosis. Nevertheless, we also provided transgenic reporter evidence for the reverse regulatory relationship, namely that *PGC1α* is required for *ATGL* expression. It is therefore possible that there is bidirectional positive regulation between ATGL and PGC1α, which is important for mitochondrial function and compromised by exposure to HFD. We also showed that boosting ATGL activity increases *ATGL* reporter expression, thus suggesting the existence of an ATGL positive feedback loop. However, additional evidence is required before it can be determined whether the observed *ATGL* reporter effects are specific for ATGL transcription or reflect more general changes in gene expression at either the transcriptional or posttranscriptional levels. Although the complete molecular pathways accounting for how increased ATGL expression in HFD nephrocytes rescues PGC1α-dependent mitochondrial processes are not yet known, it is plausible that ATGL activates transcription factors cooperating with the PGC1α coactivator. For example, it has been reported that mammalian ATGL releases lipolytic products that can activate the nuclear receptor PPARα, a partner of PGC1α, either directly or via the Sirtuin 1 deacetylase [61,62]. Another non-mutually exclusive possibility is that ATGL could rescue HFD nephrocytes via channelling fatty acids from lipid droplets into mitochondria for ß-oxidation, as has been suggested in cultured cells subject to nutrient deprivation or fatty acid toxicity [57,63]. In the context of CKD, it is known that ß-oxidation is down-regulated, and the pharmacological reversal of this has been proposed as a potential treatment [64]. Our findings now raise the possibility that pharmacological activators of lipid droplet lipolysis could be a useful addition to existing treatments for CKD. For example, small-molecule ligands for a potent activator of ATGL can boost lipolysis in adipose and muscle tissue, and it has been argued that they might be developed into therapeutic entities for obesity and diabetes [65]. In the context of CKD, our nephrocyte study suggests that it will be important to test whether this approach has the ability to induce enough beneficial lipolysis in renal cells to deal with the concomitant increase in lipid overflow from adipose tissue. If this is the case, then therapies boosting lipid droplet lipolysis could provide a novel strategy for targeting obesity-associated CKD as well as its comorbidities.

## Materials and methods

### *Drosophila* strains

Control *Drosophila* strains used in this study, including controls for Gal4/UAS experiments, were a *Wolbachia*-negative derivative of the $w^{1118}$ $iso^{31}$ strain [66] or, where specified, *UAS-mCherry RNAi* ($y^1$ $sc^*$ $v^1$ $sev^{21}$; P{y[+t7.7] v[+t1.8] = VALIUM20-mCherry}attP2. BDSC:35785). Nephrocyte and fat body–specific manipulations were performed using *Dot-GAL4* [67] and *Lpp-GAL4* [68], respectively. The following UAS fly stocks were used in this study and previously validated in the associated references: *UAS-Cubn*[i] ($w^{1118}$; P{GD6458} v14613. VDRC:v14613) [69], *UAS-mdy[i]* (P{KK102899}VIE-260B. VDRC:v100003) [55],

*UAS-bmm* [45], *Rab7::YFPmyc* [40], *UAS-LDAH::eGFP* (*UAS-CG9186::eGFP*) [38], and *UAS--Delg[i]* (*y¹ v¹; P{y[+t7.7] v[+t1.8] = TRiP.JF01805}attP2*. BDSC:25795). Similar results were obtained using *UAS-Srl[i]* (*P{KK100201}VIE-260B*. VDRC: v103355) [54] or *UAS-Srl[i]* (*y¹ sc* v¹ sev²¹; P{y[+t7.7] v[+t1.8] = TRiP.HMS00858}attP2*. BDSC:33915) [70].

## Standard and high-fat diet, larval staging, and PQQ treatment

All stocks were raised on our STD at 25˚C unless otherwise stated. STD contains 58.5 g/L glucose, 66.3 g/L cornmeal, 23.4 g/L dried yeast, 7.02 g/L agar, 1.95 g/L Nipagen, and 7.8 mg/L Bavistin, unless specified otherwise. Flies were left to lay eggs for 2 hours, on plates containing grape juice agar with yeast paste in the centre. After egg maturation for 24 hours, hatched L1 larvae were collected from the agar plates during a 1-hour time window using blunt forceps, and 20 to 25 individuals transferred to each vial at 25˚C with the appropriate diet and raised to wandering L3 stage (approximately 90 hours after larval hatching) for nephrocyte analysis. HFD corresponds to STD supplemented with 20 mM oleic acid and PQQ was added to the diet at 0.3 mM.

## Immunostaining and confocal microscopy

Larvae were inverted and fixed in 4% PFA in PBS for 30 minutes at 25˚C. After fixation, samples were washed 3 times in PBS and dissected further, if necessary. Samples were blocked in 10% normal goat serum (NGS) in PBS + 0.2% Triton (PBT), incubated overnight at 4˚C with primary antibodies diluted in 10% NGS in PBT, washed 3 times in PBT over 1 hour, incubated overnight at 4˚C with secondary antibodies diluted in 10% NGS in PBT, and then washed 3 times in PBT over 1 hour. Primary antibodies used were chicken anti-GFP at 1:1,000 (Abcam, ab13970), rat anti-KDEL 10C3 at 1:300 (Abcam, ab12223), and mouse anti-ATP5A 15H4C4 at 1:100 (Abcam, ab14748); the secondary antibodies were Alexa Fluor conjugated antibodies (Thermo Fisher Scientific, Waltham, Massachusetts, U.S.A) used at concentration 1:500. For neutral lipid staining, larvae were inverted in PBS and fixed overnight at 4˚C in 2% PFA in PBL (75 mM lysine, 37 mM sodium phosphate buffer (PB) at pH 7.4). Pericardial nephrocytes were dissected further in PBS, permeabilized for 4 minutes in 0.1% PBT, washed 3 times for 10 minutes in PBS, and stained with LipidTox Deep Red o/n at 4˚C. All samples were mounted in Vectashield (Vector Laboratories, Burlingame, California, U.S.A). For volume measurements, samples were mounted in a well generated by 1 layer of magic tape (Scotch) to avoid compression. All samples were imaged on a Leica SP5 upright microscope (Leica Microsystems, Wetzlar, Germany) using oil immersion objectives. Samples for direct quantitative comparison were imaged on the same day using the same settings. For volume measurements, confocal Z stacks spanning the entire depth of the tissue were acquired (step size of 1 um), and analyses were carried out using the procedure to identify objects inside other objects involving the Internalize Objects task of Volocity v6 (Quorum Technologies, Puslinch, Ontario, Canada).

## Ex vivo nephrocyte uptake assays

Dextran uptake assays were performed as described [31] with some modifications. Wandering L3 larvae were inverted in Schneider's Insect Medium, excess tissue was removed, and larval carcasses with CNS and pericardial nephrocytes attached were incubated for 30 minutes at 25˚C in Schneider's Medium with 10 kDa AlexaFluor568-dextran and 500 kDa FITC-dextran at a concentration of 0.33 mg/ml. For albumin uptake assay, pericardial nephrocytes were incubated for 30 minutes at 25˚C in Schneider's Medium (S0146, Merck, Darmstadt, Germany) with FITC-albumin and Red DQ-albumin at a concentration of 0.1 mg/ml. Next, tissues were washed with ice-cold PBS and fixed with 4% formaldehyde for 20 minutes at room temperature (RT). If neutral lipid staining was required, tissues were permeabilised with 0.1%

PBT for 5 minutes at RT, washed extensively with PBS, and stained with LipidTox 633 o/n at 4˚C. For BODIPY FL C12 (Thermo Fisher Scientific, D3822) uptake assays, pericardial nephrocytes were incubated for 30 minutes at 25˚C in Schneider's Medium with 0.5 mg/ml delipidated BSA (A9205, Merck) and 10 μM BODIPY FL C12 green fluorescent fatty acid. Tissues were then washed with ice-cold PBS and fixed with 4% formaldehyde for 20 minutes at 25˚C. Stained tissues were mounted in Vectashield on glass slides with a coverslip spacer of 1 layer of Scotch tape and imaged on a Leica SP5 as described above. The mean fluorescence intensity of nephrocytes (manually selected as the regions of interest) was calculated from maximum projections of Z stacks using the open-source Fiji plug-in Multi Measure.

## Lipid peroxidation assay

To detect lipid peroxidation in nephrocytes, nephrocytes were dissected in Schneider's Medium and incubated for 30 minutes in Schneider's Medium containing 10% NGS and 2 μM BODIPY 581/591 C11 (D3861 Invitrogen, Waltham, Massachusetts, U.S.A). Samples were washed and mounted in Schneider's Medium, then control and experimental samples were imaged sequentially for the non-oxidised (excitation: 561 nm, emission: 570 to 610 nm) and oxidised (excitation: 488 nm, emission: 500 to 540 nm) forms. The oxidised:non-oxidised ratio was measured in each nephrocyte, and intensity modulated ratiometric images were generated using Volocity v6 (Quorum Technologies).

## GC–MS assay for hemolymph fatty acids

Hemolymph was extracted from 10 L3 larvae (90 to 94 hours after larval hatching, ALH) using a workflow adapted from Fernando and colleagues [71]. In brief, larval groups were weighed in a small plastic dish on a microbalance and opened with forceps in a 12.5-μl drop of NaCl/13C-formate solution. A total of 7.5 μl of the hemolymph extract were transferred to 200 μl of reverse-osmosis deionised water (containing a sodium-4,4-dimethyl-4-silapentane-1-sulfonate standard) and filtered through a 0.22-μm filter unit. Moreover, 200 μl of flow-through were then transferred to 750 μl of 2:1 methanol:chloroform in a glass vial, where polar and apolar phases were separated by the addition of 250 μl of chloroform (containing a lauric acid-1-$^{13}$C internal standard) and 25 μl of water. A total of 400 μl of apolar phase was then transferred to glass vial inserts and dried down in a vacuum concentrator. The dry apolar samples were redissolved in 25 μl of 2:1 chloroform:methanol and 5 μl of the derivatisation agent, trimethylphenylammonium hydroxide solution (79266 Sigma-Aldrich, Darmstadt, Germany). Analysis of derivatised apolar metabolites was performed using an Agilent 7890A-7000C GS-MS system (Agilent Technologies, Santa Clara, California, U.S.A) operating in electron ionisation (EI) mode. A sample of 1 μl was injected into a deactivated splitless liner (270˚C) using helium as the carrier gas (0.9 ml/min) and transferred to a 30 m × 0.25 mm DB-5MS + 10 m DuraGuard capillary column. The initial oven temperature was 70˚C (1 minute), then ramped up to 230˚C (15˚C per minute), then to 325˚C (at 25˚C per minute, then a 3-minute hold). Data analysis was performed using an in-house MATLAB script (MANIC) adapted from the GAVIN package [72]. Fatty acid methyl-esters (FAMEs) were identified by comparison to authentic standards. Raw ion counts of hemolymph FAMEs were normalised to the internal standard and to larval group wet weights.

## Electron microscopy

For CLEM, dextran uptake assays were performed on dorsal vessel–pericardial nephrocyte complexes from STD and HFD larvae as described above. After washing in cold PBS, tissues were fixed with 4% paraformaldehyde in PB for 1 hour and flat mounted in 1.5% low melting temperature agarose in PB on glass coverslips. Rab7::YFP$^{myc}$ expressing nephrocytes were

imaged on a Zeiss LSM880 Airyscan confocal microscope (Carl Zeiss AG, Oberkochen, Germany) using a 63 × 1.4 NA oil immersion objective. Z stacks were obtained at 0.5-μm step size using Auto Z Brightness Correction. Airyscan processing was performed using default settings in the ZEN software. Samples were then removed from the glass slide, and excess agarose was trimmed off. They were then postfixed with 2% paraformaldehyde and 2.5% glutaraldehyde for 1 hour, prior to SBF SEM.

For SBF SEM, nephrocytes dissected from *Dot>DGAT1[i]* and *Dot>ATGL* larvae were first subjected to dextran uptake assays, and representative cells then selected for SBF SEM analysis. Nephrocytes were then fixed with 2% or 4% paraformaldehyde and 2.5% glutaraldehyde in PB for 1 hour, washed in PB, and flat mounted in 1.5% low melting temperature agarose in PB on glass coverslips. All samples, including those for CLEM, were processed for SBF SEM using the previously described protocol with modifications [73]. Briefly, tissues were postfixed in 2% osmium tetroxide and 1.5% potassium ferricyanide for 1 hour, incubated in 1% thiocarbohydrazide for 20 minutes, followed by 2% osmium tetroxide for 30 minutes. Osmicated tissues were then stained en bloc with 1% uranyl acetate overnight, followed by Walton's lead aspartate staining for 30 minutes at 40 to 60˚C. Tissues were then dehydrated with a graded ethanol series, flat-embedded in Durcupan ACM resin, and polymerised at 60˚C. Samples were mounted onto aluminum pins using conductive epoxy glue (ITW Chemtronics, Kennesaw, Georgia, U.S.A) and trimmed to the region of interest guided by light microscopy images. Trimmed blocks were sputter-coated with 5- to 10-nm platinum using a Q150R S sputter coater (Quorum Technologies). SBF SEM data were collected using a 3View2XP (Gatan, Pleasanton, California, United States of America) attached to a Sigma VP SEM (Zeiss, Cambridge, United Kingdom). The microscope was operated at 2.0 to 2.3 kV with 30-μm aperture, using Variable Pressure mode or Focal Charge Compensation mode [74]. Inverted backscattered electron images were acquired through entire nephrocytes every 50 or 100 nm, at a resolution of 6.5 to 8.0 nm/pixel. Acquired images were imported into Fiji [75] and aligned using the Register Virtual Stack Slices [76]. For the S1–S4 Movies, aligned data were scaled down to 50 nm/pixel and encoded into the H. 264 compression format using the Fiji/ImageJ plugin imagej-ffmpeg-recorder.

## SBF SEM quantification of endosomes

CLEM was used to define the morphology of the endocytic compartments to be quantified using SBF SEM. The Airyscan Z stack was matched to the SBF SEM data using the Fiji plugin BigWarp. Endolysosomes were manually selected in each stack as landmarks, and thin-plate spline transformation was applied to match the 2 stacks. Approximately 70 endolysosomes were classified using CLEM into 5 morphology groups based on their SEM luminal density and their Dextran and Rab7::YFP[myc] status in the corresponding Airyscan images (**S3A Fig**). Dark endolysosomes with a luminal density similar to or higher than the cytoplasm were all Dextran[−] (but Rab7[+] and Rab7[−]), and therefore, along with Golgi apparatus associated vesicles, were not segmented in SBF SEM images. White and light endosomes were segmented on 1 or more SBF SEM slices including the midplane of the compartment by fitting the largest inscribed circle using the Fiji plugin TrakEM2 [77]. Then using the Fiji 3D Object Counter [78], the size of the bounding box was used to estimate the object diameter, and this was used to calculate the spherical volume. For quantitation of endosome numbers and volumes, a size threshold of 300-nm diameter was selected and validated by showing that it gave comparable endosome size distributions for the different fixation protocols used for CLEM or for standard SBF SEM (**S3B Fig**).

## Statistical analysis

R version 3.5.1 (July 2, 2018) was used for all statistical analysis (R Core Team, 2018). Boxplots were generated using ggplot2, show the median with first and third quartile, and whiskers extend from the hinge by 1.5× interquartile range. Data points are coloured according to which independent experiment they are from. For statistical analyses, data were modelled using a linear model (LM), a linear mixed model (LMM), or a general linear mixed model (GLMM) with diet as fixed and independent experiment as random effect followed by a Wald chi-squared test. Asterisks show statistical significance (* $p < 0.05$, ** $p < 0.005$, *** $p < 0.0005$). Data were modelled using restricted maximum likelihood (REML), LMMs, or GLMMs from the lme4 package [79]. The model fit was evaluated using normal quantile–quantile plots. Experimental manipulation such as diet, genetic manipulation, and dextran size were categorised as fixed effects, and independent experiments were categorised as random effects. Statistical inference for fixed effects was tested by Wald chi-squared test from the R car package [80]. For multiple comparisons, estimated marginal means (EMMs) were predicted using the R emmeans package [81] and comparisons used Bonferroni correction. Statistical methods, parameters, and results for each figure are summarised in **S1 Data**, and the corresponding source data are provided in **S2 Data**.

## Supporting information

**S1 Fig. HFD and genetic lipid overflow models do not disrupt growth and developmental timing.** (**A**) Hemolymph abundances of myristic (C14:0), myristoleic (C14:1), and oleic (C18:1) acids in STD and HFD larvae. Abundance (ion counts per mg) was measured by GC–MS and normalised to larval wet weights. (**B–D**) Graphs compare STD and HFD animals, indicating that they have similar larval weight (mg), nephrocyte volume (μm³), and developmental timing (% pupariation versus hours after larval hatching). Note that nephrocyte size is significantly different ($p < 0.0005$) between STD and HFD animals. (**E–G**) Graphs compare STD animals expressing ATGL in the fat body (*Lpp>ATGL*) with controls (*Lpp-GAL4*). indicating that they have similar larval weight (mg), nephrocyte volume (μm³), and developmental timing (% pupariation versus hours after larval hatching). Note that nephrocyte size is significantly different ($p < 0.0005$) between control and ATGL expressing animals. See **S1 Data** for details of *p*-values and the type of statistical model used for all graphs in this study. **S2 Data** provides the source data used for all graphs and statistical analyses. ATGL, adipose triglyceride lipase; GC–MS, gas chromatography–mass spectrometry; HFD, high-fat diet; STD, standard diet.
(TIF)

**S2 Fig. Time course of dextran uptake in nephrocytes.** (**A**, **B**) Graphs quantify individual uptake (A) and uptake ratio (B) of fluorescently labelled 500-kDa and 10-kDa dextrans as a function of time (min) for ex vivo pericardial nephrocytes. See **S1 Data** for details of *p*-values and the type of statistical model used for all graphs in this study. **S2 Data** provides the source data used for all graphs and statistical analyses.
(TIF)

**S3 Fig. CLEM analysis of endolysosomes in STD nephrocytes.** (**A**) The 5 endolysosomal categories distinguished in CLEM analysis of *Rab7*::*YFP^myc^* STD nephrocytes subjected to dextran uptake assays. The criteria used were scanning EM luminal density ("white," "light," or "dark"), and also the "+" or "−" status of expression of Dextran and Rab7::GFP. Scale bar = 1 μm. (**B**) Quantitations from SBF SEM volumes of a STD control and a STD *Rab7*::*YFP^myc^* nephrocyte showing similar endosome size distributions. Note that "white" and "light"

endosomes but not "dark" endolysosomes were segmented and quantified per cell according to their diameter (µm). See **S1 Data** for details of *p*-values and the type of statistical model used for all graphs in this study. **S2 Data** provides the source data used for all graphs and statistical analyses. CLEM, correlative light-electron microscopy; SBF SEM, serial blockface scanning electron microscopy; STD, standard diet.
(TIF)

**S4 Fig. DGAT1 knockdown increases nephrocyte lipid peroxidation.** (**A**) Confocal panels represent ratio of oxidised (500- to 540-nm emission) to non-oxidised (570- to 610-nm emission) forms of the lipid peroxidation sensor BODIPY 581/591 C11 in pericardial nephrocytes (dotted outlines) from STD larvae and HFD larvae carrying *Dot>DGAT1[i]* or *Dot>ATGL*. (**B**) Graph quantifies oxidised:non-oxidised ratios of the lipid peroxidation sensor BODIPY 581/591 C11 in pericardial nephrocytes for the dietary and genetic manipulations in A. On HFD, lipid peroxidation is increased by *DGAT1* knockdown but not by *ATGL* expression. See **S1 Data** for details of *p*-values and the type of statistical model used for all graphs in this study. **S2 Data** provides the source data used for all graphs and statistical analyses. ATGL, adipose triglyceride lipase; DGAT1, diglyceride acyltransferase 1; HFD, high-fat diet; STD, standard diet.
(TIF)

**S5 Fig. *Srl* knockdown prevents ATGL rescue of dextran uptake.** (**A**) *Delg* or *Srl* knockdown decreases mitochondrial volume in STD nephrocytes. Quantitation of mitochondrial volumes (as % of cell volume) of STD pericardial nephrocytes for control (*Dot>ctrl*), *Dot>Delg[i]* and *Dot> Srl[i]* larvae. (**B**) Confocal panels show 10-kDa and 500-kDa dextran uptake in ex vivo pericardial nephrocytes of control (*Dot>ctrl*) and *Dot>ATGL; Srl[i]* larvae on HFD. The dextran signals of both genotypes, imaged within the same field of view, are comparable. Graph shows small but significant decrease in 10-kDa and 500-kDa dextran uptake between control (*Dot>ctrl*) and *Dot>ATGL; Srl[i]* larvae on HFD. Note that in the absence of Srl knockdown, ATGL significantly increases dextran uptake on HFD. See **S1 Data** for details of *p*-values and the type of statistical model used for all graphs in this study. **S2 Data** provides the source data used for all graphs and statistical analyses. ATGL, adipose triglyceride lipase; HFD, high-fat diet; Srl, Spargel; STD, standard diet.
(TIF)

**S1 Data. Summary of statistical methods and analysis.** For each main and supporting figures, the linear mixed models, statistical inference tests, and *p*-values are shown.
(XLSX)

**S2 Data. Source data.** The numerical values used for all main and supporting figures and for the statistical analyses in S1 Data are provided.
(XLSX)

**S1 Movie. SBF SEM Z stack of a STD nephrocyte.** Data were collected with a 0.2-µm step size and recorded at 20 fps. SBF SEM, serial blockface scanning electron microscopy; STD, standard diet.
(MOV)

**S2 Movie. SBF SEM Z stack of a HFD nephrocyte.** Data were collected with a 0.2-µm step size and recorded at 20 fps. HFD, high-fat diet; SBF SEM, serial blockface scanning electron microscopy.
(MOV)

**S3 Movie. SBF SEM Z stack of a *Dot>DGAT1[i]* HFD nephrocyte.** Data were collected with a 0.2-μm step size and recorded at 20 fps. DGAT1, diglyceride acyltransferase 1; HFD, high-fat diet; SBF SEM, serial blockface scanning electron microscopy.
(MOV)

**S4 Movie. SBF SEM Z stack of a *Dot>ATGL* HFD nephrocyte.** Data were collected with a 0.2-μm step size and recorded at 20 fps. ATGL, adipose triglyceride lipase; HFD, high-fat diet; SBF SEM, serial blockface scanning electron microscopy.
(MOV)

## Acknowledgments

We acknowledge Mathias Beller, R. Kühnlein, and the late Susan Abmayr and Suzanne Eaton for fly stocks and antibodies. Fly stocks were also obtained from the Bloomington *Drosophila* Stock Center (NIH P40OD018537), the Vienna Drosophila Research Centre, and the Kyoto *Drosophila* Genetic Resource. We thank the Crick Metabolomics STP, Clare Newell and Ian McGough for assistance with experiments as well as for helpful advice and discussions. We also thank Eva Islimye and Elisabeth Kamper for comments on the manuscript.

For the purpose of open access, the author has applied a CC BY public copyright licence to any Author Accepted Manuscript version arising from this submission.

## Author Contributions

**Conceptualization:** Aleksandra Lubojemska, M. Irina Stefana, Alex P. Gould.

**Formal analysis:** Aleksandra Lubojemska, Sebastian Sorge, Andrew P. Bailey, Lena Lampe, Azumi Yoshimura, Alana Burrell, Alex P. Gould.

**Funding acquisition:** Alex P. Gould.

**Investigation:** Aleksandra Lubojemska, M. Irina Stefana, Sebastian Sorge, Andrew P. Bailey, Azumi Yoshimura.

**Methodology:** Aleksandra Lubojemska, M. Irina Stefana, Sebastian Sorge, Andrew P. Bailey, Azumi Yoshimura, Lucy Collinson, Alex P. Gould.

**Resources:** Lucy Collinson, Alex P. Gould.

**Supervision:** M. Irina Stefana, Lucy Collinson, Alex P. Gould.

**Validation:** Aleksandra Lubojemska, Lena Lampe, Azumi Yoshimura.

**Visualization:** Alex P. Gould.

**Writing – original draft:** Aleksandra Lubojemska, Alex P. Gould.

**Writing – review & editing:** Aleksandra Lubojemska, M. Irina Stefana, Sebastian Sorge, Andrew P. Bailey, Lena Lampe, Azumi Yoshimura, Lucy Collinson, Alex P. Gould.

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
