## [Editor Report · Decision Letter 0]

6 Oct 2020

Dear Alex, 

Thank you for submitting your manuscript entitled "Adipose Triglyceride Lipase protects the endocytosis of renal cells on a high fat diet in Drosophila" for consideration as a Research Article by PLOS Biology.

Your manuscript has now been evaluated by the PLOS Biology editorial staff as well as by an academic editor with relevant expertise and I am writing to let you know that we would like to send your submission out for external peer review.

Please re-submit your manuscript within two working days, i.e. by Oct 08 2020 11:59PM.

Best wishes,

Ines

--

Ines Alvarez-Garcia, PhD

Senior Editor

PLOS Biology

---

## [Decision Letter · Decision Letter 1]

17 Nov 2020

Dear Alex,

Thank you very much for submitting your manuscript entitled "Adipose Triglyceride Lipase protects the endocytosis of renal cells on a high fat diet in Drosophila" for consideration as a Research Article at PLOS Biology. Thank you also for your patience as we completed our editorial process, and please accept my apologies for the delay in providing you with our decision. Your manuscript has been evaluated by the PLOS Biology editors, an Academic Editor with relevant expertise, and by two independent reviewers.

You will see that both reviewers are very enthusiastic about your work, however they also raise several issues that need to be addressed. While Reviewer 2 only has two minor points, Reviewer 1 asks for a long list of requests, but only two are deemed to be essential and require experiments: showing that free fatty acid, or DAG in the hemolymph is involved in the tissue communication (Point 7) and whether the effects on bmm are at the level of transcriptional or post transcriptional regulation (Point 13). After discussing these requests with the academic editor, we do not think Point 13 is essential for publication, but you would need to soften the discussion of the results to include the possibility of post-transcriptional regulation.

In light of the reviews (attached below), we will not be able to accept the current version of the manuscript, but we would welcome re-submission of a revised version that takes into account the reviewers' comments. We cannot make any decision about publication until we have seen the revised manuscript and your response to the reviewers' comments. Your revised manuscript is also likely to be sent for further evaluation by the reviewers.

We expect to receive your revised manuscript within 3 months. 

**IMPORTANT - SUBMITTING YOUR REVISION**

*Re-submission Checklist*

*Published Peer Review*

*PLOS Data Policy*

*Blot and Gel Data Policy*

Best wishes,

Ines

--

Ines Alvarez-Garcia, PhD,

Senior Editor,

ialvarez-garcia@plos.org,

PLOS Biology

Reviewers’ comments

Rev. 1:

The study by Lubojemska and colleagues starts out with developing a Drosophila nephrocyte-based model for mammalian chronic kidney disease (CKD). Nephrocytes of fly larva on high fat diet accumulate lipid droplets and also recapitulate several other aspects of CKD including ER and mitochondrial deficits as well as compromised endocytosis. ATGL/bmm overexpression in the fat body of larvae on standard diet causes similar phenotypes. Tissue-specific impairment of TAG synthesis or excessive TAG mobilization clears LDs but has very different consequences on nephrocyte function. Only ATGL/bmm overexpression rescues the adverse effects of HFD. In a final set of experiments the authors implicate ATGL/bmm transcriptional control by HFD. Also, they demonstrate a role of mitochondrial regulators PGC1alpha/Spargel and Delg in HFD-induced nephrocyte dysfunction.

The outline of this highly interesting and well-written study is clear, the experiments are technically sound and - with exceptions - provide convincing evidence for the conclusions drawn. The CLEM studies are impressive, and the statistical treatment of the data is excellent. The figures are clear and informative. Yet, the readers` desire to understand the mechanistic connections tying the correlative phenotypes together are not fully satisfied.

As outlined below, the experiments concerning the transcriptional control of ATGL/bmm are not convincing. A further major drawback is the lack of any direct evidence for the hemolymph lipids (FFAs?, DAG?) the authors imply to cause the pathophysiological effects in nephrocytes in larva on HFD or in larva on normal food, which over-express ATGL/bmm in the adipose tissue.

The elegant genetic experiments leave of lot of space for interpretations concerning the underlying lipid-based mechanisms, which cause nephrocyte malfunction on HFD. The possible consequences of DAGT1/mdy knockdown deserve a broader discussion given the central role of DAG as signaling lipid and intermediate of phospholipid metabolism.

Comments (in the order of appearance):

1) Comparing the mammalian and fly renal systems in Fig. 1A is informative. But various structures and cell types referred to in the text are missing and need to be added. It also remains unclear how a discontinuous nephrocyte/Malpighian tubule system works compared to a nephron.

2) Authors state that fat-body specific ATGL expression did not substantially alter nephrocyte size (line 207/208). Yet, the provided data (Fig. S1E) indicate a ~25% decrease in cell volume upon ATGL expression. This should be stated in the main text as it remains a formal possibility that decreased endocytic uptake occurs secondary to decreased cell volume.

3) Along these lines: How is nephrocyte cell volume affected by Dot-Gal4-mediated DGAT1 knockdown or ATGL overexpression specifically in nephrocytes?

4) Fig. 2A: There seems to be no correlation between dextran uptake and lipid loading. Why?

5) Fig. 2C: There seems to be no lipid loading in response to HFD. Why?

6) Fig. 3 refers to Rab7::GFP not YFP.

7) ESSENTIAL FOR REVISION: The authors invoke but never show increased hemolymph free fatty acid (or DAG, as in fact the BODIPY C12 experiments argue against a contribution of free fatty acids) to be involved in the tissue communication between adipose tissue and nephrocytes on HFD or upon adipose tissue ATGL/bmm overexpression on STD. This needs to be experimentally shown.

8) Is there a formal prove that Lpp-Gal4 does not drive in nephrocytes?

9) What does "Lpp> " and "Dot> " refer to? Driver line only or crossed to (which?) control?

10) Line 208 (S1D-S1F Fig.) needs attention.

11) The two left panels in Fig. S4 show signal of (non-oxidized) BODIPY 581/591 C11 outside the cells in a nephrocytes preparation i.e. extracellularly. How is this possible?

12) Line 301/2: Statement on the HFD-dependent regulation of ATGL/bmm needs support of a reference.

13) ESSENTIAL FOR REVISION: Line 304: What is the evidence for a transcriptional regulation? Post-transcriptional regulation or translational control of GFP protein in the physiologically compromised nephrocytes from HFD animals could plausibly account for differences in fluorescence intensity. Fig. 6A-D data are not providing any convincing evidence and need to be complemented by additional experiments addressing transcriptional regulation more directly.

14) Line 316: Introduce GABPA

15) Fig. 5 (significant difference) and 6E (not difference) data on Dot>ctrl and Dot>DGAT[i] are contradictory. Why? This difference questions the effect of PQQ in DGAT knockdown conditions.

16) Fig. 5A appears to show a fuzzy neutral lipid staining in Dot-ATGL nephrocytes. Why is that?

17) Why does mitochondrial cell volume of Dot-ATGL differ by about 5% between 5C/6E and 6F? This is relevant as it is the unusual high Dot-ATGL value in 6F, which makes Delg[i] and Srl[i] significantly different.

18) Figure 5 (and elsewhere): How can the authors be sure that reversal of the ATGL-overexpression effect upon concomitant DGAT1 or Srl knockdown is not due to a simple GAL4 dilution effect? According to the figure caption, ATGL was not co-expressed with ctrl RNAi, which would have been a more appropriate setup.

19) Does Srl[i] and Delg[i] also block LD clearance in Dot>ATGL expressing nephrocytes of HFD specimens? This would be in line with a reduction of flux through the LD compartment in cell with compromised mitochondrial function in spite of increased lipolytic capacity.

20) The authors are encouraged to add fly stock numbers for those strains received from stock centers.

21) Do nephrocytes accumulate LDs under starvation, when ATGL/bmm gets physiologically up-regulated in the adipose tissue?

22) Does DGAT1/mdy overexpression in nephrocytes of HFD larva increase LD size/number and protect against HFD-induced phenotypes mitochondrial and endocytosis phenotypes? This finding would strengthen the argument that lipid flux into the LD compartment acts protective.

23) Comment on the SI: the supplementary information is informative and complements the main text/figures

Rev. 2: Barry Denholm – this reviewer has waived anonymity

In this paper the authors demonstrate a high fat diet (HFD) induces lipid droplet formation/accumulation in nephrocytes. They go on to show the HFD results in nephrocyte phenotypes including a substantial reduction in ER and mitochondrial volumes. These data reveal strong phenotypic similarities between the nephrocyte model and podocyte/proximal tubule in animals/patients on HFDs. These data suggest the nephrocyte is a powerful model to dissect the role of lipids in pathogenesis of CKD.

Nephrocytes are prodigious endocytosers. The authors go on to show these endocytic functions are highly attenuated by a HFD by assaying for endocytosed material and by quantifying endosome number. Further, evidence is provided that lipid droplets might be caused from overflow from fat body and subsequent take-up by nephrocytes. Take-up in nephrocytes is shown to be Cubulin-dependent.

The authors go on to elegantly dissect the question of the contribution of lipid droplets on HFD-induced renal dysfunction using a combination of cell-type specific knock-down/activation and epistatic studies. They show that ATGL expression but not DEGAT1 knock-down is sufficient to rescue HFD-induced phenotypes in nephrocytes, and go on to define the pathway of the ATGL rescue.

The data is comprehensive, strong and supports the interpretations made. Overall, I thought the approach was very elegant and the manuscript very well argued. I would strongly recommend publication in your journal. I don't suggest further experiments.

Given the high prevalence of CKD and a lack of understanding into the contribution of lipotoxicity in the disease it is likely to be of wide interest to readers of PLoS. The authors describe a new model to study HFD-induced CKD and use cell-type specific genetic manipulations, epitasis experiments and pharmacology to illustrate how powerful this model can be in revealing disease pathways/mechanisms.

Two minor points the authors might choose to consider:

(1) The authors comment briefly on changes to nephrocyte size: an increase in response to HFD (line 88, S1B); and a decrease in Lpp>ATGL animals (line 208, S1E). Both changes are significant and might influence quantification of parameters measured in the experiments. Did the authors account for this? Perhaps an absolute count of lipid droplet number would be useful to include alongside (if cell size is scaling independently in these experiments).

(2) Might it be interesting to consider correlation between lipid droplets and phenotypes (for e.g. mitochondrial and ER organelle reduction) on a cell-by-cell basis, i.e. do more droplets mean stronger phenotype?

---

## [Decision Letter · Decision Letter 2]

23 Mar 2021

Dear Alex,

Thank you for submitting your revised Research Article entitled "Adipose Triglyceride Lipase protects the endocytosis of renal cells on a high fat diet in Drosophila" for publication in PLOS Biology. I have now obtained advice from one of the original reviewers and have discussed these comments with the Academic Editor. 

Based on the reviews, we will probably accept this manuscript for publication, provided you satisfactorily address the data and other policy-related requests.

We expect to receive your revised manuscript within two weeks. 

-  a cover letter that should detail your responses to any editorial requests.

*Published Peer Review History*

*Early Version*

Best wishes,

Ines

--

Ines Alvarez-Garcia, PhD,

Senior Editor,

ialvarez-garcia@plos.org,

PLOS Biology

Fig. 1C, E; Fig. 2B, D; Fig. 3B; Fig.4B, D, F, H; Fig. 5A-H; Fig. 6B-G; Fig. S1A-G; Fig. S2A, B; Fig. S3B; Fig. S4B and Fig. 5A, B

BLURB

Please also provide a blurb which (if accepted) will be included in our weekly and monthly Electronic Table of Contents, sent out to readers of PLOS Biology, and may be used to promote your article in social media. The blurb should be about 30-40 words long and is subject to editorial changes. It should, without exaggeration, entice people to read your manuscript. It should not be redundant with the title and should not contain acronyms or abbreviations. For examples, view our author guidelines: https://journals.plos.org/plosbiology/s/revising-your-manuscript#loc-blurb

Reviewer's comments

Rev. 1:

All the relevant points of concern were satisfayingly address. I congratulate the authors to a very intersting study.

---

## [Editor Report · Decision Letter 3]

13 Apr 2021

Dear Alex,

On behalf of my colleagues and the Academic Editor, Emma Rawlins, I am pleased to say that we can in principle offer to publish your Research Article entitled "Adipose Triglyceride Lipase protects renal cell endocytosis in a Drosophila dietary model of chronic kidney disease" in PLOS Biology, provided you address any remaining formatting and reporting issues. These will be detailed in an email that will follow this letter and that you will usually receive within 2-3 business days, during which time no action is required from you. Please note that we will not be able to formally accept your manuscript and schedule it for publication until you have made the required changes.

PRESS

Thank you again for supporting Open Access publishing. We look forward to publishing your paper in PLOS Biology. 

Best wishes,

Ines

--

Ines Alvarez-Garcia, PhD 

Senior Editor 

PLOS Biology